# Mowing Facilitated Shoot and Root Litter Decomposition Compared with Grazing

**DOI:** 10.3390/plants11070846

**Published:** 2022-03-23

**Authors:** Shuzhen Zhang, Yuqi Wei, Nan Liu, Yongqi Wang, Asiya Manlike, Yingjun Zhang, Bo Zhang

**Affiliations:** 1College of Grassland Science and Technology, China Agricultural University, Beijing 100193, China; xjauzsz98@gmail.com (S.Z.); yuqi.wei@zalf.de (Y.W.); liunan@cau.edu.cn (N.L.); 2College of Grassland Science, Xinjiang Agricultural University, Urumqi 830052, China; xjauwyq97@gmail.com; 3Pratacultural Research Institute of Xinjiang Academy of Animal sciences, Urumqi 830052, China; hwyj@xjaas.net

**Keywords:** mountain steppe ecosystems, grazing, mowing, altitude, litter decomposition

## Abstract

Shoot and root litter are two major sources of soil organic carbon, and their decomposition is a crucial nutrient cycling process in the ecosystem. Altitude and land use could affect litter decomposition by changing the environment in mountain grassland ecosystems. However, few studies have investigated the effects of land use on litter decomposition in different altitudes. We examined how land-use type (mowing vs. grazing) affected shoot and root litter decomposition of a dominant grass (*Bromus inermis*) in mountain grasslands with two different altitudes in northwest China. Litterbags with 6 g of shoot or root were fixed in the plots to decompose for one year. The mass loss rate of the litter, and the environmental attributes related to decomposition, were measured. Litter decomposed faster in mowing than grazing plots, resulting from the higher plant cover and soil moisture but lower bulk density, which might promote soil microbial activities. Increased altitude promoted litter decomposition, and was positively correlated with soil moisture, soil organic carbon (SOC), and β-xylosidase activity. Our results highlight the diverse influences of land-use type on litter decomposition in different altitudes. The positive effects of mowing on shoot decomposition were stronger in lower than higher altitude compared to grazing due to the stronger responses of the plant (e.g., litter and aboveground biomass) and soil (e.g., soil moisture, soil bulk density, and SOC). Soil nutrients (e.g., SOC and soil total nitrogen) seemed to play essential roles in root decomposition, which was increased in mowing plots at lower altitude and vice versa at higher altitude. Therefore, grazing significantly decreased root mass loss at higher altitude, but slightly increased at lower altitude compared to mowing. Our results indicated that the land use might variously regulate the innate differences of the plant and edaphic conditions along an altitude gradient, exerting complex impacts in litter decomposition and further influencing carbon and nutrient cycling in mountain grasslands.

## 1. Introduction

Litter decomposition plays an essential role in regulating soil organic carbon (SOC) formation, nutrient cycling, atmospheric carbon dioxide concentration, and plant growth [1,2], as about 50% of the aboveground net primary productivity returns to the soil as shoot litter [3]. Fine root litter from belowground is also an important source of SOC and nutrients [4], which might play a more critical role than shoot litter [5]. Litter decomposition is governed by the litter quality, decomposer community, and the physical and chemical characteristics of the environment [6,7]. The soil physicochemical factors, such as soil temperature, moisture, and pH, predominately affect litter decomposition by regulating soil microbial activity [8,9]. However, how those factors affect decomposition varies with the scale at which they are examined [10]. Therefore, determining the pathways controlling shoot and root litter decomposition is the basis of quantitative analysis of the carbon and nitrogen cycle in terrestrial ecosystems [11,12], especially following changes in climate and human activities.

The grassland ecosystem, which covers 40% of global land, is an essential barrier for ecological security and a foundation for pastoral production [13]. Grazing and mowing are two of the most critical land-use types in grasslands [11,14]. They could alter plant nutrient content and stoichiometry [15], and soil properties, including soil micro-communities and their activities [8], thus affecting litter decomposition [16,17]. Mowing reduces the aboveground plant inputs to soil [18,19], while grazing presents more complex effects on grasslands [20,21]. Grazing decreases the plant coverage and increases soil temperature, resulting in a soil moisture reduction and a concomitant slower litter decomposition [22]. Furthermore, trampling increases soil bulk density and reduces soil humidity and aeration, restricting microbial activities and litter decomposition [23,24]. Moreover, 60–90% of the nutrients ingested by livestock are returned to the soil in the form of dung and urine [25]. The dung and urine provide high-quality nutrients for microorganisms, improving microbial activities and promoting litter decomposition [26]. However, there is no clear consensus on the effects of land-use types on litter decomposition in grassland ecosystems [27].

The temperature and precipitation change along the altitudinal gradient forms specific soils and vegetation [28,29]. Therefore, the altitude might influence litter decomposition through the direct effects of altitude-dependent climates and indirect impact from various soil and vegetation conditions [30]. However, there are no consistent results on how the litter decomposition changes along the altitude gradient, with both faster [31,32] and lower [33] decomposition rates in higher than lower altitude reported in previous research. The differences in moisture and temperature are the main factors affecting microbial activities and litter decomposition along an altitude gradient [34,35]. Generally, the temperature decreases, and the activities of litter-decomposers are accordingly reduced, resulting in lower litter decomposition with an increase in altitude [33], especially in cold areas where the mean annual temperature (MAT) is lower than 6.75 °C [7]. However, some studies have showed that moisture, but not temperature, was the key limiting factor affecting litter decomposition [36,37,38]. Therefore, the primary limitation of altitude on litter decomposition varies among ecosystems [39]. Moreover, the innate differences in ecosystems along an altitude gradient might modify the responses of litter decomposition to land-use types (e.g., mowing vs. grazing) [30]. For instance, the decreased soil moisture when suffering grazing might be mitigated in high-altitude conditions. Few studies have investigated the interactions between altitude and land use on the environmental conditions and the litter decomposition in the mountain ecosystems.

This study explored the effects of land-use type and altitude on the shoot and root litter decomposition in a mountain grassland ecosystem. We aim to assess how the changes in vegetation and soil conditions induced by altitude and land-use type affect litter decomposition and determine the main controlling factors. We hypothesize that (1) shoot and root litter will decompose faster under mowing than grazing where microenvironment favors decomposition (e.g., greater soil moisture and lower bulk density); (2) an increase in altitude will decline litter and root decomposition with lower temperature; (3) altitude and land-use type may have interactive effects on the microenvironment, shoot and root decomposition, and their relationships.

## 2. Results

### 2.1. Effects of Altitude and Land-Use Type on Shoot and Root Decomposition

The shoot and root mass loss respectively ranged from 40% to 59% and 35% to 69% after decomposing for one year (Figure 1). The shoot and root mass loss were significantly greater under mowing than grazing (*p* < 0.001), and also under higher than lower altitude (*p* < 0.01, Figure 1). Moreover, land-use type effects on the shoot and root decomposition varied between lower and higher altitude (*p* < 0.05) (Figure 1). Specifically, in lower altitude, grazing restricted shoot decomposition but promoted root decomposition with a slight response (shoot: grazing 39.68% vs. mowing 46.13%, −13.98%; root: grazing 40.64% vs. mowing 35.09%, +15.82%). Grazing resulted in much more shoot (grazing 40.48% vs. mowing 58.99%, −31.38%) and root mass loss (grazing 40.64% vs. mowing 69.30%, −41.36%) than mowing at a higher altitude. The negative effects of grazing offset the positive effects of increasing altitude on decomposition; therefore, mass loss in grazing plots between the two altitudes was similar. The main effects of altitude resulted from mowing plots (Figure 1).

### 2.2. Effects of Altitude and Land-Use Type on Soil Properties and Plant Attributes

The effects of altitude and land-use type on soil temperature and moisture were opposite. Soil temperature declined while moisture increased from lower to higher altitude, and grazing increased soil temperature while reducing moisture compared with mowing (*p* < 0.001, Figure 2A,B). There was a marginally significant interaction of altitude and land-use type on soil moisture, where the grazing-induced reduction in moisture is greater in higher (−11.53%) than lower (−6.19%) altitude (*p* < 0.1, Figure 2B). Soil bulk density was lower in higher sites, and grazing caused higher bulk density than mowing (*p* < 0.01). Moreover, the grazing effect on bulk density was strengthened in higher than lower sites (*p* < 0.1, +79.60% vs. +24.28%, Figure 2C). There were greater SOC and soil TN at higher altitude (*p* < 0.01), and similar TN was detected in mowing and grazing plots (*p* > 0.05, Figure 2D and Appendix A). The influences of land-use type on SOC varied between altitudes, with greater SOC in grazing at lower altitude, but less at higher altitude (*p* < 0.001, Figure 2D).

The activity of β-xylosidase in the shoot and root was greater at higher than lower altitude, but was only significant for the shoot (*p* < 0.05, Figure 2E,F). Moreover, the activity of β-xylosidase in the shoot (*p* < 0.05) and root (*p* = 0.05) showed different responses to land-use type. At lower altitude, the activity of β-xylosidase was greater under grazing than mowing, while the opposite was true at higher altitude (Figure 2E,F).

Plant communities in mowing plots showed more litter and aboveground biomass compared with grazing plots (*p* < 0.01, Figure 2H and Appendix A), and this difference was stronger at higher than lower altitudes (*p* < 0.05, Figure 2H and Appendix A). Litter and aboveground biomass declined with the increasing altitude in grazing, while the opposite occurred in mowing plots (*p* < 0.1, Figure 2G and Appendix A). In addition, *B. inermis* was the dominant species under mowing in the two altitudes (Appendix A). A poisonous grass, *Achnatherum inebrians*, appeared in the grazing area, and occurred more in lower than higher altitude grazing plots (important value, low altitude 0.45 vs. high altitude 0.08) (Appendix A). There was greater plant coverage in higher than lower altitude, and in mowing than grazing treatments (*p* < 0.001, Figure 2H).

### 2.3. The Relationships between the Environment Variables and Decomposition of Shoot and Root

The correlation heatmap showed that the shoot and root mass loss had significant positive correlations with plant coverage, litter biomass, soil moisture, and SOC, and was significantly negatively correlated with soil temperature and soil bulk density (*p* < 0.05, Figure 3). Root mass loss was also positively related to the root β-xylosidase activity (Figure 3). The RDA analysis showed that differences in plant and soil properties could explain 70.94% of the variation in shoot and root decomposition (*p* < 0.01, Figure 4). Besides, the β-xylosidase activity of the litter produced the same trend with soil moisture and SOC, and the opposite trend with soil temperature and bulk density (*p* < 0.05, except not significant for the bulk density and shoot β-xylosidase activity, Figure 3 and Figure 4). The change in soil moisture is positively related to plant coverage and negatively related to soil temperature and bulk density (*p* < 0.05, Figure 3 and Figure 4). Moreover, litter decomposition performed a stronger relationship with plant coverage and litter biomass, while root decomposition was closer to SOC (Figure 4).

## 3. Discussion

This study showed that altitude and land-use type significantly influenced plant and soil properties, resulting in shoot and root decomposition differences. Consistent with our first hypothesis, we found greater plant coverage, litter biomass, soil moisture, and SOC, but lower bulk density in the mowing than the grazing treatment, which led to faster litter decomposition. The effect of altitude was the opposite of our second hypothesis. We found that an increase in altitude promoted litter decomposition accompanied by lower bulk density and greater soil moisture and activities of β-xylosidase. Altitude and land-use type interactively affected shoot and root decomposition as predicted by the last hypothesis; specifically, the land-use type showed more significant impacts on the environmental conditions and decomposition of shoot and root in the higher than lower altitude. Our results indicated that grasslands along a vertical distribution respond differently to changes in utilization, which might regulate ecosystem functions, e.g., litter decomposition. These findings are helpful to reveal the C cycling and the management of grassland ecosystems [16,17,19].

We found faster decomposition of shoot and root at higher altitude, which accompanies changes in plant and soil properties, and is in line with previous results [31]. The effects of altitude on decomposition depend on the critical limiting factor [39], which seems to be soil moisture but not the temperature in this research and is consistent with previous studies [31,37,38,40]. Although, it was reported that the temperature decreased and accordingly reduced the activities of litter decomposers with the rise in altitude, resulting in lower litter decomposition [33]. In contrast, litter decomposed faster in our study with greater soil moisture at higher altitude. The decline in soil bulk density and rise in SOC indicated stronger microbial activities supported by greater β-xylosidase activity in higher than lower altitude, contributing to litter decomposition [41,42]. All the above changes might counteract the negative effects of decreasing soil temperature. It is worth mentioning that only two altitudes were determined in this study as restricted by the limited study areas, which might lead to different results compared with previous research. Previous research reported that there was a threshold of water limitation on plant growth along wide altitude gradients [43,44], which indicated the positive effects of an increase in soil moisture might be outweighed by a decrease in temperature in ever-rising altitudes.

Overall, mowing promoted greater shoot and root decomposition compared with grazing. The decay of litter also changed synchronously with plant and soil properties. Mowing was conducted in September, while livestock continually consumed aboveground biomass in grazing plots. This brought about higher plant coverage and litter biomass during the growing season following mowing compared to grazing. Less canopy following reduced plant coverage and litter accumulation provided less shade in grazing plots, which increased soil temperature and induced lower soil moisture by evaporation, as supported by our results and previous research [22,45]. Moreover, mammal trampling increased soil bulk density, which might induce soil compaction and salinization, restricting microbial activity and litter decomposition [21]. Grazing effects on litter decomposition depend on grazing strategies (e.g., grazing intensities, grazing season, etc.) and vary among different decomposition stages [46,47]. Recent research found that light grazing promoted litter decomposition, while heavy grazing tended to reduce or show limited effects on litter decomposition compared with grazing exclusion [47]. This research could support our results as the intensity in the study areas is heavy grazing.

The effects of land-use type on the shoot and root decomposition varied between the two altitudes, which were stronger in high rather than low sites. The positive effects of mowing on shoot decomposition were stronger at higher than lower altitude compared to grazing, while grazing dramatically decreased root mass loss at high altitude, but slightly increased at low altitude compared to mowing. Compared with low altitude, there were relatively stronger reductions in soil moisture and SOC, and an increase in soil bulk density following grazing in high altitude, which would amplify the negative grazing effects on litter decomposition. The changes in soil properties might result from the plant growth and their C inputs. Furthermore, poisonous grass, *Achnatherum inebrians*, appeared following heavy grazing and was rarely taken by animals in the grasslands [48]. It indicated that the growth of this poisonous grass could contribute to the accumulation of litter. The number of *A. inebrians* seeding also decreased with altitude [49], which resulted in more *A. inebrians* at low than high altitudes (important value, low altitude 0.45 vs. high altitude 0.08). Therefore, the plant and litter biomass responded more strongly to grazing at high altitudes and resulted in a greater reduction in soil moisture and litter mass loss. These changes in plant biomass and litter indicated plant C inputs to soil are varied along an altitude gradient. The nutrients returned in the form of dung and urine in grazing plots were supposed to supply SOC and microbial activity, which was only confirmed by the results from low altitude with sufficient plant C inputs. The greater SOC seemed to be critical for faster root decomposition in grazing than mowing plots at low altitude, as we found a strong relationship between SOC and root mass loss. On the contrary, the greater decrease in aboveground biomass and litter following grazing in high than low altitudes indicated fewer C inputs to soil, which might further limit microbial activity and litter decomposition [26,50]. As discussed above, the effect of land use on decomposition could be regulated by grazing strategies in grazing grasslands, and the influence of altitude on decomposition might vary among broader gradients. Therefore, future research targeting litter decomposition under different intensities of land uses along a broader altitude gradient is required to further improve our understanding of nutrient cycling in mountain grasslands.

## 4. Materials and Methods

### 4.1. Study Site

The study site was in the Grassland Ecological Research Station of Xinjiang Agriculture University, Xinjiang, China (43°28′ N, 87°01′ E), located in the middle of the Northern Slope of Tianshan Mountain. This site has a semiarid continental monsoon climate with annual precipitation of 300–400 mm, annual evaporation of 1100–1300 mm, annual mean temperature of 3–4 °C, and a frost-free period of about 130 days. The soil is mountain chestnut soil and the soil properties (0–200 mm) are: soil organic matter = 30.1 ± 0.13 g kg^−1^, the total nitrogen = 0.40 ± 0.05 g kg^−1^, the total phosphorus = 0.35 ± 0.03 g kg^−1^, pH = 8.37 ± 0.07. During the 1970s and 1980s, the degraded grasslands were reseeded with cultivated grass species, including *Bromus inermis* and *Onobrychis viciifolia*, forming a stable plant community. The dominant plant species are *B. inermis*, *Elytrigia repens*, *Achillea millefolium*, *Vicia sepium, Thalictrum aquilegiifolium*, *Medicago falcata*, *O. viciifolia*, *Taraxacum mongolicum*, and *Potentilla anserina* [51]. In this study area, the average annual temperature decreases by 0.3–0.4 °C [52], and the precipitation increases by 20–25 mm every 100 m above sea level [53]. These differences allow us to investigate the effects of altitude on litter decomposition.

### 4.2. Experimental Design

The experimental plots were set up at altitudes of 1600 m (hereafter lower altitude) and 1800 m (hereafter higher altitude). In the past two decades, mowing and grazing have been the primary land-use types at low and high altitudes in this region. For grazing areas, livestock include cattle and sheep, from May to June and August to September every year, with a grazing intensity of 2.4–2.8 standard sheep units hm^−2^ yr^−1^ (sheep with a weight of 50 kg as a standard sheep unit (SSU) [54,55]). For the mowing area, livestock was excluded by the fence, and mowing was conducted with sickles by native herdsmen (about 100 mm stubble) once a year in early September. Therefore, there were four treatments with five replications each: low altitude + mowing, low altitude + grazing, high altitude + mowing, high altitude + grazing (Figure 5). In 2019, twenty 10 m × 10 m plots were established, randomly extending to both sides of the exclosure fence that separated grazing and mowing areas at each altitude. The distances between the plots were set as more than 10 m to reduce the autocorrelation between plots [56]. Previous research reported that plant communities exhibited spatial autocorrelation over a range of about 2 m in grasslands, which indicated that the plots in the same treatment area in this study should be independent of each other [57].

The standing shoot and roots of *B. inermis* were collected from the fenced area in the research station in October 2018. To ensure uniformity in the litter, we only collected standing litter with three leaves and similar lengths. At the same place, we dug the roots in the 0–100 mm soil layer, washed them with water, and picked out the fine roots (<2 mm). All plant materials were dried at 80 °C for 48 h.

Shoot and root decomposition were studied using the litterbag method [19]. The weighed shoot and root (6 g) were put into nylon mesh bags (200 mm × 150 mm, 1 mm pore). On 1 June 2019, we put four litterbags in each plot, including two litter and two root bags. In total, eighty litterbags: two altitudes × two land-use types × two decomposition materials × five replications × two parallel determinations (one for mass loss and another for the activity of β-xylosidase, more details in Section 4.3) in each plot, were deployed in the twenty plots. The aboveground litterbags were fixed with 80 mm nails after removing other litter on the ground. The root bags were buried into the 0–100 mm soil layer.

### 4.3. Litter Sampling

After the shoot and root decomposed for one year, litterbags were collected on 1 June 2020. All bags were removed from each plot and immediately placed into dry ice to prevent further decomposition and preserve extracellular enzymes. Litterbags were transported to the lab and stored at −80 °C. Any visible fresh vegetation soil particles adhering to the shoot or root were removed, and the remaining materials were over-dried at 80 °C for 48 h and weighed. The dry matter loss ratios of litter and root were calculated according to the followed equation [45].
*Mass loss* (%) = (*M*_0_ − *M_t_*)/*M*_0_ × 100
where *M*_0_ is the initial weight and *M_t_* is the residual amount after the decomposition time of *t*.

We assayed the activity of β-xylosidase using pNP-β-xyloside. For the extraction of β-xylosidase activities, a 10% *w/v* pulp homogenate was prepared by homogenizing 0.1 g of freeze-dried pulp tissue in an extraction buffer prepared according to previous work [58]. β-xylosidase activity assay was undertaken in the crude extract as described in Wei et al. [59]. The enzyme activity was calculated in μmol per gram per hour using the standard equations outlined by DeForest [60].

### 4.4. Plant and Soil Sampling and Measurements

In August 2019, one quadrat (1 m × 1 m) was randomly set in each plot to determine the plant coverage and aboveground biomass. The plant coverage was estimated visually in each quadrat plot. All the plants in the quadrat were cut, dried at 80 °C for 48 h, and then weighed.

After removing the litter layer, ten soil cores (25 mm in diameter) were randomly collected from the top 100 mm layer and mixed into one sample in each plot. Soil temperature was measured with a geothermometer (Dongfang Huabo (Beijing) Technology Co., Ltd., Beijing, China; M404121). Soil moisture was determined by the drying method [19]. Soil bulk density was determined from undisturbed soil samples collected using a core sampler [61]. Sediment slurries (10 g dry soil and 50 mL deionized water shaken for one hour, 250 rpm) were used to measure soil pH [62]. Soil organic carbon (SOC) was determined by Walkley–Black wet oxidation method [63], and total nitrogen (TN) was determined using the Kjeldahl method [64].

### 4.5. Statistical Analysis

Generalized linear models were conducted to test the effects of altitude and land-use type on all the variables. Normality and equal variance assumptions were tested by the Kolmogorov–Smirov test and the Levene test, respectively. Log transformation was applied when necessary to improve the homogeneity of variance. The generalized linear models were carried out with SPSS18.0, and the visualization was conducted with an R package of ggplot2 (R 3.6.0). The correlation coefficients (Pearson R^2^) and correlation tests (*p* < 0.05) were employed to evaluate the relationships between all the variables. The correlation analysis and the visualization were conducted with R packages of *correlation* and *corrplot*, respectively (R 3.6.0). The redundancy analysis (RDA) was used to explore further how the litter decomposition was influenced by environmental variables and the relationships between environmental variables (R 3.6.0, R package of *vegan*).

## 5. Conclusions

The present study found that altitude and land-use type had significant interactive effects on the shoot and root decomposition in montane steppe ecosystems. Altitude influenced shoot and root mass loss by changing soil moisture, pH, and soil organic carbon; root decomposition was also affected by β-xylosidase activity. Grazing reduced soil moisture and SOC, and increased soil bulk density, slowing down the decomposition of shoot and root compared with mowing. Moreover, the microenvironment responded more to the land-use type in higher than lower altitudes, contributing to stronger shoot and root decomposition effects. Our results indicated that, compared with free grazing, mowing benefits litter accumulation, litter decomposition, and nutrient cycling in mountain grasslands, especially in higher altitude. Hence, an improved understanding of how altitude and grassland utilization affect shoot and/or root decomposition might help us to better manage the mountain grassland steppe ecosystems. However, further research is needed to study the effects of different land-use intensities on litter decomposition along broader altitude gradients, and conclude a complete understanding of nutrient cycling in mountain grasslands with anthropogenic disturbance.

## Figures and Tables

**Figure 1 plants-11-00846-f001:**
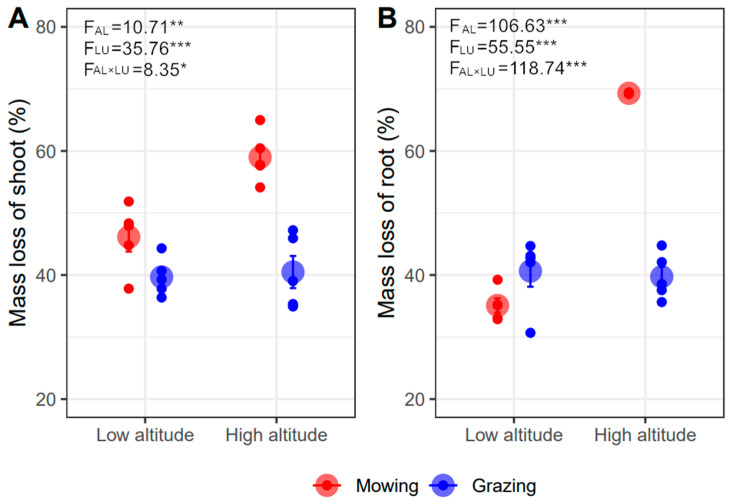
Effects of altitude (AL) and land-use type (LU) on shoot (**A**) and root (**B**) mass loss after one year of decomposition (Mean ± SE). The *, **, and *** represent *p* < 0.05, *p* < 0.01, and *p* < 0.001.

**Figure 2 plants-11-00846-f002:**
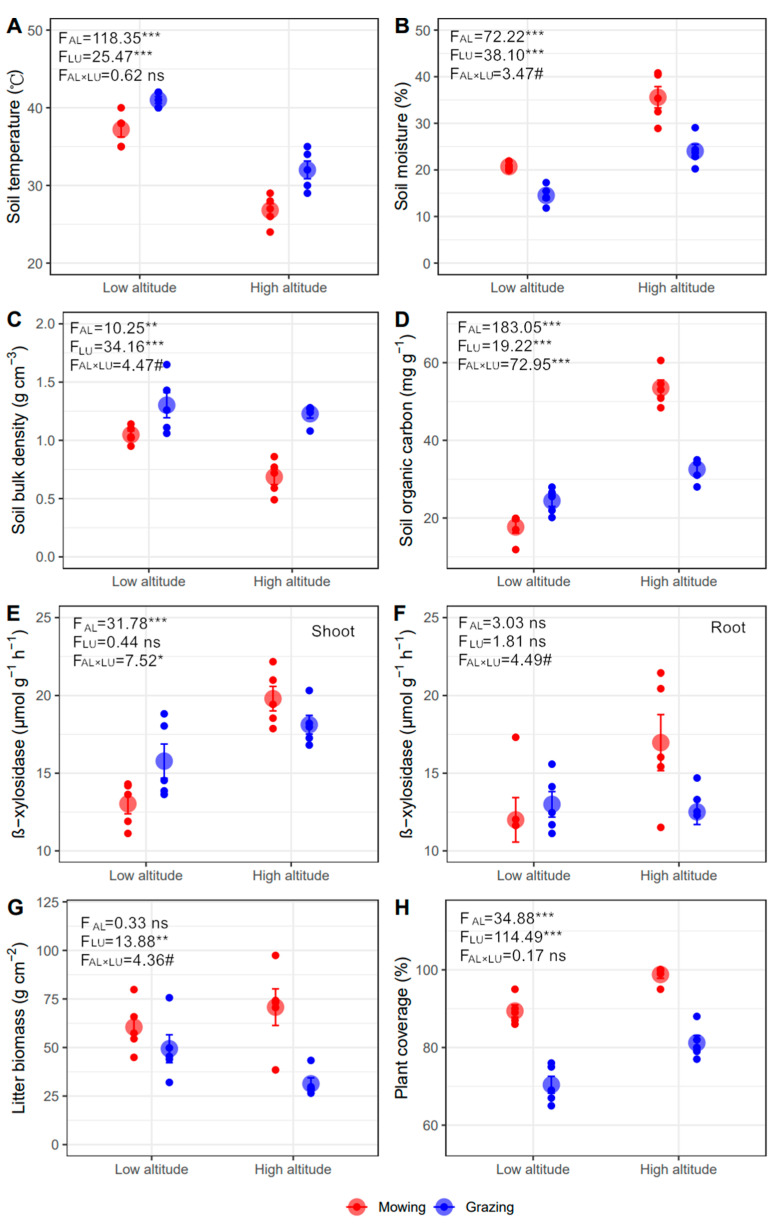
Soil and plant properties under different land-use types at two altitudes (mean ± SE). (**A**) soil temperature, (**B**) soil moisture, (**C**) soil bulk density, (**D**) soil organic carbon (SOC), (**E**) β-xylosidase of shoot litter, (**F**) β-xylosidase of root litter, (**G**) plant coverage, and (**H**) litter biomass. The ns, #, *, **, and *** represent *p* > 0.1, *p* < 0.1, *p* < 0.05, *p* < 0.01, and *p* < 0.001.

**Figure 3 plants-11-00846-f003:**
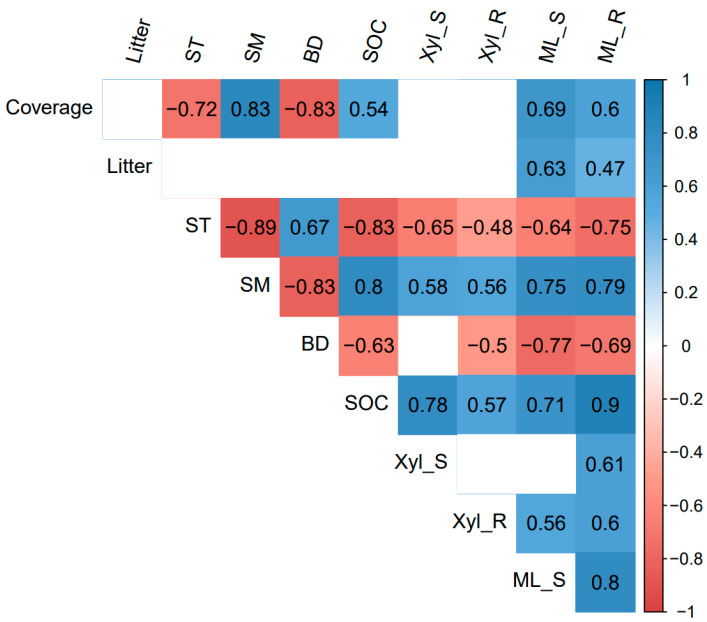
Heatmap of Pearson correlation between 10 variables. Litter: litter biomass; ST: Soil temperature; SM: Soil moisture; BD: bulk density; SOC: soil organic carbon; Xyl_S: β-xylosidase of shoot; Xyl_R: β-xylosidase of root; ML_S: mass loss of shoot; ML_R: mass loss of root. HG: high altitude + grazing; HM: high altitude + mowing; LG: low altitude + grazing; LM: low altitude + mowing. *p*-value set at 0.05 for significance.

**Figure 4 plants-11-00846-f004:**
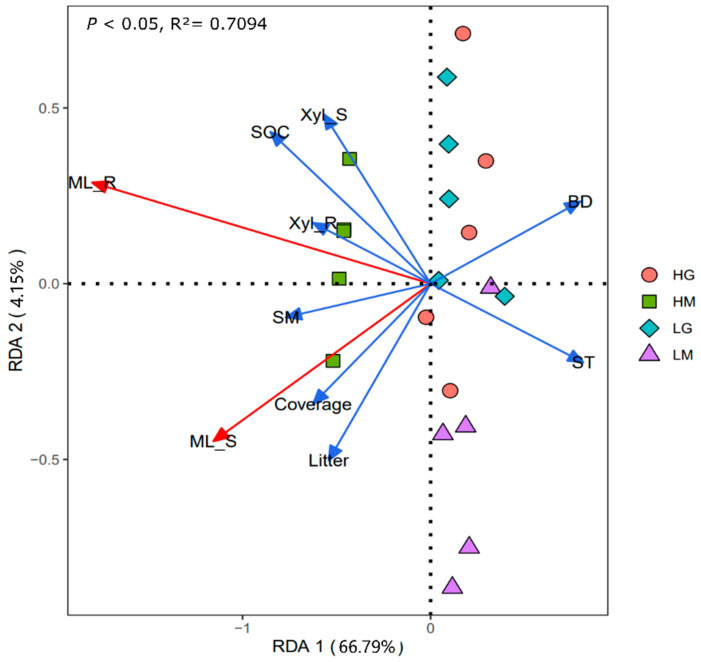
Redundancy analysis graph of the relationships between the plant and soil properties and mass loss of shoot and root (*p* < 0.05). Litter: litter biomass; ST: Soil temperature; SM: Soil moisture; BD: bulk density; SOC: soil organic carbon; Xyl_L: β-xylosidase of shoot; Xyl_R: β-xylosidase of root. HG: high altitude + grazing; HM: high altitude + mowing; LG: low altitude + grazing; LM: low altitude + mowing.

**Figure 5 plants-11-00846-f005:**
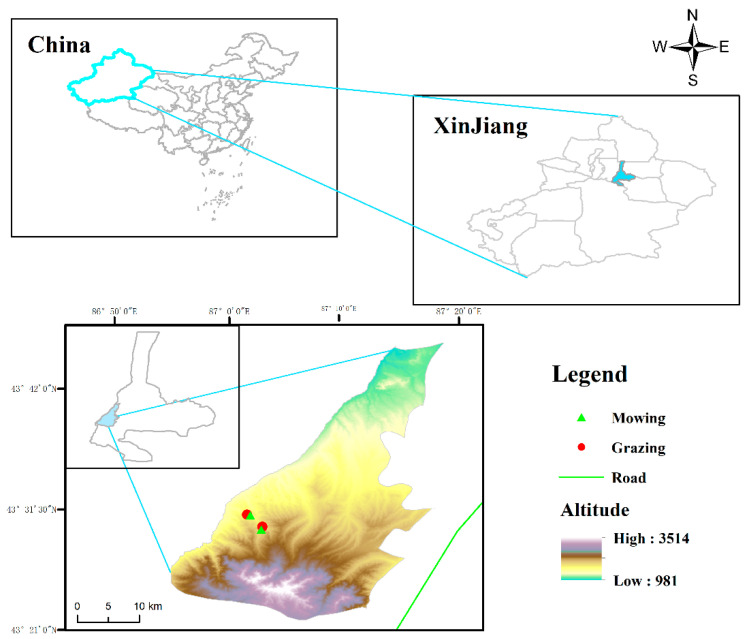
Location of the study area. The low altitude + mowing (87°01′49″ E, 43°30′58″ N; 1631 m asl), low altitude + grazing (87°01′32″ E, 43°31′02″ N; 1612 m asl), high altitude + mowing (87°02′47″ E, 43°29′44″ N; 1809 m asl), high altitude + grazing (87°02′53″ E, 43°29′59″ N; 1804 m asl).

## Data Availability

The data presented in this study are available on request from the corresponding author. The data are not publicly available due to privacy.

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
