# Peer review of "Mowing Facilitated Shoot and Root Litter Decomposition Compared with Grazing"

_plants, 2022, doi:10.3390/plants11070846_

Round 1

Reviewer 1 Report

The manuscript deals with a relevant subject to PLANTS related with land use effects on shoot and root litter decomposition in different altitudes. The ms is very interesting, well written, with an interesting set of data and supported by deep and up-to-date literature and adequate discussion. I recommend that the manuscript should be accepted after minor revision.

Minor points:

  1. Lines 13-14: "and litter decomposition their decomposition". Correct.
  2. Line 76: spell out MAT.
  3. Authors should provide information about how mowing was done.
  4. Line 247: lack of comma between treatments.
  5. Line 265: 4.3 instead of 2.3.

Author Response

To: editors and reviewers,

From: Yingjun Zhang, Bo Zhang, Shuzhen Zhang, Yuqi Wei,

zhangyj@cau.edu.cn, xjauzb@126.com

Subject: Major Revisions requested plants_1643953

Dear editors and reviewers,
Attached, please find a revised version of Manuscript ID plants_1643953 entitled " Mowing facilitated shoot and root litter decomposition compared with grazing, especially at high altitude" which incorporates reviewer recommendations.

Thank you so much for taking time to review our manuscript! Your comments and suggestions helped us greatly improve the clarity and value of our manuscript. As suggested, we have carefully evaluated each comment and revised our manuscript accordingly (the changes we made are in red text). We have provided a point-by-point description of our response to reviewers’ comments below. Our responses to reviewers’ suggestions are in red text.

Response to Reviewer 1’s Comments

1. Lines 13-14: "and litter decomposition their decomposition". Correct.
We have corrected ‘and litter decomposition their decomposition’ to ‘their decomposition’ (see line 13).

2. Line 76: spell out MAT.

We have added the full spelling of ‘MAT’ in line 75. Thank you for this suggestion.

3. Authors should provide information about how mowing was done.

We have added how mowing was done in lines 275-276.

4. Line 247: lack of comma between treatments.

We have added a comma between treatments (see line 277).

5. Line 265: 4.3 instead of 2.3.

Thank you for your suggestion. We have replaced 2.3 with 4.3 (see line 295).

Thank you again for the thorough and extremely helpful reviews! We would be glad to respond to any further questions or comments you may have on this revision.

Sincerely,

Yingjun Zhang, Bo Zhang, Shuzhen Zhang, Yuqi Wei

Department of Grassland Science, College of Animal Science and Technology

China Agricultural University, Beijing CHINA

Tel: +86-10-62733380;

Email: zhangyj@cau.edu.cn, xjauzb@126.com

Reviewer 2 Report

The paper's topic is relatively narrow. I appreciate the large amount of experimental work. I have a few key comments about this article:

It is not appropriate to state the main conclusion in the paper's title. It needs to be adjusted.

The order in which the chapters are presented is a significant factor. Listing Material and Methods as chapter 4 is confusing and illogical. Please adjust 1. Introduction; 2. Material and Methods; 3. Results; 4. Discussion; 5. Conclusions.

The terms high or low altitude have different values in different parts of the world. Please edit throughout the text.

Introduction:

  • Missing data on the species spectrum of mountain pasture and meadow vegetation
  • Define scientific hypotheses and, if necessary, set sub-objectives

Results:

  • Determining the influence of altitude based on only two locations is not possible. Adjustments are required.
  • Grazing values will be more affected by the animals' load and intensity of grazing. It needs to be adjusted
  • You completely ignore vegetation species composition and competition between plant species. This needs to be added.

Material and Methods:

  • Use SI units. Replace "cm" with "m" or "mm"
  • It is not clear according to which system you use botanical plant names
  • Precisely define "sheep unit · hm-2"

Conclusions:

  • Based on your methodology, it is not possible to determine the effect of altitude. Your results only show a condition that occurs at two different altitudes.
  • It is unclear how important your results are for the mountain pasture ecosystem; what are the risks and benefits.

English errors:

Line 177 - … but not the temperature in this research and consistent.. – It seems that you are missing a verb – is consistent

Author Response

To: editors and reviewers,

From: Yingjun Zhang, Bo Zhang, Shuzhen Zhang, Yuqi Wei,

zhangyj@cau.edu.cn, xjauzb@126.com

Subject: Major Revisions requested plants_1643953

Dear editors and reviewers,
Attached, please find a revised version of Manuscript ID plants_1643953 entitled " Mowing facilitated shoot and root litter decomposition compared with grazing, especially at high altitude" which incorporates reviewer recommendations.

Thank you so much for taking time to review our manuscript! Your comments and suggestions helped us greatly improve the clarity and value of our manuscript. As suggested, we have carefully evaluated each comment and revised our manuscript accordingly (the changes we made are in red text). We have provided a point-by-point description of our response to reviewers’ comments below. Our responses to reviewers’ suggestions are in red text.

1. It is not appropriate to state the main conclusion in the paper's title. It needs to be adjusted
Thank you for your suggestion. We changed the title to “Mowing facilitated shoot and root litter decomposition compared with grazing in a mountain grassland” for consideration.

2. The order in which the chapters are presented is a significant factor. Listing Material and Methods as chapter 4 is confusing and illogical. Please adjust 1. Introduction; 2. Material and Methods; 3. Results; 4. Discussion; 5. Conclusions.

Yes, the structure is different from some other journals, but we prepared this ms according to the MDPI LaTeX templates. We are open to the final structure of this ms if you are insistent on the regular one you mentioned above.

3. The terms high or low altitude have different values in different parts of the world. Please edit throughout the text.

Thank you for pointing out our need to clarify this concern. We agree that the terms high or low altitude could be context-dependence. To improve the clarity and accuracy, we emphasize the high and low altitude as two different altitudes. Therefore, we changed our statement, e.g., using “higher or lower” instead of “high or low”. Please see details in Lines 16, 18, 25, 26, 29-30, 70, 97-100, 104-106, 113, 117-123, 125-129, 132, 136, 138, 141, 181, 186, 193, 195, 269-270, 349.

4. Introduction:

o   Missing data on the species spectrum of mountain pasture and meadow vegetation

Sorry, we are not sure if we get you correctly. Do you mean we need to give the species composition in our research areas? If it’s true, please see it in the section “Materials and Methods” (Lines 257-260). If it’s not, would you please give some more specific information on your concern? Thank you!

o   Define scientific hypotheses and, if necessary, set sub-objectives

Thanks for the suggestion, and we added a paragraph to describe our objectives and give scientific hypotheses. Lines 84-92.

5. Results:

o   Determining the influence of altitude based on only two locations is not possible. Adjustments are required.

Thank you for pointing out the concern. We agree that it might be insufficient to determine the effects of altitude based on only two levels, but we restricted by the study areas as it is not easy to find so paired grazing and mowing grasslands along altitudes. It will decrease the generalization of our results. However, we already see significant different soil and plant properties between those two different altitudes and significant influence of altitude on litter and root decomposition even with two altitudes. We tried to emphasize our results within two altitudes instead of altitudes gradient (all through the main text). We added some sentences to address this concern in the section Discussion; please see Lines 196-202, 241-246.

o   Grazing values will be more affected by the animals' load and intensity of grazing. It needs to be adjusted

We agree with your concern and add some sentences in Discussion (Lines 214-219, 241-246) to address the effects of grazing strategies on litter decomposition.

o   You completely ignore vegetation species composition and competition between plant species. This needs to be added.

Thank you for your helpful and meticulous suggestions. We presented the results of dominant species composition (Lines 134-139 and Table S1) and proportion of important values of various species (Lines 226-231 and Figure S3). The influence of vegetation species composition on litter decomposition was indirectly via its effects on litter accumulation and microenvironments, so we decided to show it in the supplemental materials.

6. Material and Methods:

o   Use SI units. Replace "cm" with "m" or "mm"

Thank you for your suggestions. We have replaced ‘cm’ with ‘mm’ (see Lines 254, 276, 288, 291, 296-298, 321-322).

o   It is not clear according to which system you use botanical plant names

Thank you for pointing out this concern. We used the binomial nomenclature to name plants, we have verified and revised the plant names in Lines 258-259.

o   Precisely define "sheep unit · hm-2"

Thank you for pointing out this concern. We have supplemented the definition of the standard sheep unit and the references to the source of the definition (see Lines 273-274).

7. Conclusions:

o   Based on your methodology, it is not possible to determine the effect of altitude. Your results only show a condition that occurs at two different altitudes.

As mentioned above, we have tried to emphasize the difference between the two altitudes instead of an altitude gradient and point out the limitations and further efforts needed in the future (see Lines 354-357).

o   It is unclear how important your results are for the mountain pasture ecosystem; what are the risks and benefits.

We have added some sentences to address the importance of our results (see Lines 350-354).

8. English errors:

Line 177 - … but not the temperature in this research and consistent. – It seems that you are missing a verb – is consistent

Revised; please see line 189.

Thank you again for the thorough and extremely helpful reviews! We would be glad to respond to any further questions or comments you may have on this revision.

Sincerely,

Yingjun Zhang, Bo Zhang, Shuzhen Zhang, Yuqi Wei

Department of Grassland Science, College of Animal Science and Technology

China Agricultural University, Beijing CHINA

Tel: +86-10-62733380;

Email: zhangyj@cau.edu.cn, xjauzb@126.com

Round 2

Reviewer 2 Report

Thank you for your cooperation